# sPLA2 Wobbles on the Lipid Bilayer between Three Positions, Each Involved in the Hydrolysis Process

**DOI:** 10.3390/toxins14100669

**Published:** 2022-09-26

**Authors:** Natalia Kuzmina, Pavel Volynsky, Ivan Boldyrev, Anna Alekseeva

**Affiliations:** 1Frumkin Institute of Physical chemistry and Electrochemistry, Russian Academy of Sciences, 119071 Moscow, Russia; 2Shemyakin–Ovchinnikov Institute of Bioorganic Chemistry, Russian Academy of Sciences, 117997 Moscow, Russia

**Keywords:** PLA2, lipids, hydrolysis, AFM, supported lipid bilayers, molecular dynamics

## Abstract

Secreted phospholipases A2 (sPLA2s) are peripheral membrane enzymes that hydrolyze phospholipids in the *sn-2* position. The action of sPLA2 is associated with the work of two active sites. One, the interface binding site (IBS), is needed to bind the enzyme to the membrane surface. The other one, the catalytic site, is needed to hydrolyze the substrate. The interplay between sites, how the substrate protrudes to, and how the hydrolysis products release from, the catalytic site remains in the focus of investigations. Here, we report that bee venom PLA2 has two additional interface binding modes and enzyme activity through constant switching between three different orientations (modes of binding), only one of which is responsible for substrate uptake from the bilayer. The finding was obtained independently using atomic force microscopy and molecular dynamics. Switching between modes has biological significance: modes are steps of the enzyme moving along the membrane, product release in biological milieu, and enzyme desorption from the bilayer surface.

## 1. Introduction

Interfacial enzymes are membrane-effected proteins, and secreted phospholipase A2 (sPLA2) is one of their bright representatives [1]. sPLA2s are found in practically all types of organisms, including mammals. They are ubiquitously expressed throughout most cells and tissues, suggesting their importance in life processes, including digestion, inflammation, autoimmunity, host defense and atherosclerosis [2,3]. The sPLA2 reaction is important for signal transduction, since polyunsaturated fatty acids (e.g., arachidonic acid) and lysophospholipids released by sPLA2 can be converted to a wide variety of bioactive lipids and promote multiple deleterious processes on their own. In addition to catalytic properties, other functions are found for various representatives of sPLA2 [2,4,5,6]. Since sPLA2 is a biomarker of cardiovascular risk [7], the development of sPLA2 inhibitors and kits for sPLA2 activity measurements are of great importance [3,8,9,10].

Controlling sPLA2 activity is very challenging, and there is currently no sPLA2 inhibitor on the market, although many are in clinical trials [8]. To a large extent, the apparent rate and in vivo activity of sPLA2 are regulated by the affinity of the enzyme to the membrane interface. Given the high conservation of the catalytic site and, conversely, the variability in the interfacial binding site [11], the details of the process of enzyme interaction with the membrane seem promising for further regulation of sPLA2 activity.

Currently, research on sPLA2 is focused on finding details of the enzyme action on the atomistic level. A combination of experimental approaches and calculation methods, especially molecular dynamics simulations (MD), is used. The last provide atomistic resolution of the enzyme-bilayer-substrate system. Recently, unique hydrophobic sites of PLA2, which promote substrate specificity, were found [12], the influence of the linkage at the *sn-1* position of the lipid on hydrolysis rate was investigated [13], the precise specificity of the subsite for the oxidized fatty acid at the *sn-2* position of the phospholipid backbone for lipoprotein-associated PLA2 was discovered [14], and details of PLA2 absorption on the bilayer surface were described [15].

Atomic force microscopy (AFM) is a useful tool in the investigation of kinetics of the PLA2-catalyzed reaction [16], with the first AFM observation of enzymatic degradation of dipalmitoylphosphatidylcholine bilayer by Grandbois et al. in 1998 [17]. The advantage of direct observation with AFM reveled the important role of defects or irregular molecular packing in the phospholipid bilayer membrane in catalysis, where they act as the active sites for the enzyme [18].

sPLA2 hydrolysis includes the stages of (1) adhesion on the membrane surface, (2) binding of an individual substrate molecule, and (3) its hydrolysis. This basic mechanism is often extended with intermediate steps to adjust it to the objects under study [1,19]. The interfacial binding step is crucial to the enzymatic function of sPLA2, as catalysis in the aqueous phase is virtually nonexistent [20].

Despite a large and expanding body of knowledge regarding sPLA2 enzymes, several topics remain unclear. These include molecular mechanisms of enzymes hopping between membranes and scooting across the bilayer, product release from the enzyme and steps involved in the interfacial recognition.

Not much attention has been paid to the process of the release of hydrolysis products. The theoretical description of sPLA2 hydrolysis assumes that the reaction products remain in the membrane [1,3]. However, in real experiments, the release of reaction products, the redistribution of all membrane components, the formation of other aggregates, or the complete destruction of the membrane are often observed [21,22,23,24]. At the moment, there is no consensus on the fate of hydrolysis products.

Gaps in our understanding of sPLA2 action have motivated us to visualize the enzyme on the bilayer surface using atomic force microscopy on supported lipid bilayer and model the enzyme landing on the bilayer surface using molecular dynamics simulations. Both methods independently revealed the occurrence of three stable binding modes (different orientations) of the bee venom sPLA2 (bvPLA2) on the palmitoyloleoylphosphatidylcholine (POPC) bilayer. Below, we report details of the finding and propose the mechanism of sPLA2 movement across the bilayer and release of hydrolysis products. The tandem use of atomic force microscopy and molecular dynamics simulations gave good results, combining the data of a real experiment with multiple protein molecules and the details of a single protein at the atomic level. This combination of methods may be useful for other soft matter studies.

## 2. Results

### 2.1. Atomic Force Microscopy Results

#### 2.1.1. Supported Lipid Bilayer (SLB)

The model lipid bilayer used in the study is a lipid bilayer supported on the mica surface (SLB). While the SLB technology is broadly used, the preparation of SLB suitable for an enzymatic process requires multiple steps of adjustment. Lipids, pH, salinity, Ca^2+^ and ethylenediaminetetraacetic acid (EDTA) concentration all affect both enzyme activity and SLB performance.

We started by applying to mica liposomes prepared in standard PLA2 buffer (100 mM or 10 mM Tris-HCl, 100 mM NaCl, pH 8.5). The liposomes had extremely low sorption and did not form a bilayer on mica. When liposomes were prepared without the use of the buffer (in milliQ water), they were well adsorbed on mica and degraded to form a stable supported bilayer. However, when washed with the buffer, the bilayer was deformed, possibly due to the osmotic shock. We were able to obtain a stable and smooth supported POPC bilayer by preparing liposomes in 100 mM NaCl. In this case, liposomes readily formed a bilayer on the mica, and further exchange of a liquid above with diluted buffer (10 mM Tris-HCl, 100 mM NaCl, pH 8.5) had no effect on bilayer performance.

Depending on the lipid concentration and incubation time, we obtained different degrees of mica coverage with the bilayer (Figure 1A,D,G). These differed from each other by the amount of defects in the lipid cover.

#### 2.1.2. Enzyme Action on SLB

The SLB was washed with the Ca buffer (10 mM Tris-HCl, 100 mM NaCl, pH 8.5, 5 mM CaCl_2_). When an active enzyme was added to a bilayer with minor defects, we observed a rapid destruction of the bilayer, which began from the edges of the defects (Figure 1B,C). Given the rate of the process and the changeability of the membrane in this case, it was not possible to visualize the enzyme in this mode. We used the data to ensure the enzyme was capable of rapid hydrolysis in our experimental conditions. The process of bilayer degradation by sPLA2 under AFM conditions was studied previously [17,18,22]. It was shown that dipalmitoylphosphatidylcholine hydrolysis catalyzed by bvPLA2 occurs preferentially at the edges of pits or defects on the bilayer surface [22].

#### 2.1.3. Visualization of Enzyme Bound to the Bilayer

To visualize the enzyme on the bilayer, we used a buffer with the addition of EDTA. At high concentrations of EDTA, the enzyme tended to adsorb on the membrane boundaries and on bare mica areas, but not on the bilayer (Figure 1E,F). EDTA is able to interact with the choline group of phospho-lipids [25], and at high concentration, it may hinder an enzyme’s adsorption on the membrane surface.

Previously, it was demonstrated that hydrolysis of the bilayer by sPLA2 starts from the defects and boundaries of the bilayer [17,26]. We also observed the accumulation of bvPLA2 in these areas. High curvature and open hydrophobic areas may provoke protein sorption and subsequent catalysis.

By adjusting the conditions so as to obtain complete coverage of the mica with the bilayer and reducing the concentration of EDTA, we obtained images of the protein on the membrane (Figure 1G–I).

Fitting the size distribution of protein globules on the bilayer with Gaussian functions revealed three populations (states) (Figure 2A,C). These differed from each other by the altitude (the distance between the outer surface of the enzyme molecule and bilayer surface) and relative frequency of occurrence. Taking into account the geometric dimensions of the protein globule, it can be assumed that the population with the smallest altitude is the protein most deeply embedded in the membrane, while other populations may represent a protein that is not so deeply embedded in the bilayer or adsorbed on it by the other modes (Figure 2B). Since the population of states (probabilities) is known, the energy difference between states could be calculated according to the Boltzmann equation (see diagram in Figure 2C).

### 2.2. Molecular Dynamics Results

Coarse grain (CG) simulations were performed for six starting orientations of the enzyme relative to the bilayer (schematic representation on Figure 3 top, CG representation Appendix A). In each run, the enzyme had rapidly landed on the bilayer, and this was traced by protein–lipid contacts (see Appendix A).

Analyzing the protein–lipid contact map (Appendix A), we distinguished a unique or repetitive set of amino acids forming contact for all six dynamics. The main result from the 5 μs CG molecular dynamics simulation was that six different starting orientations of bvPLA2 had degraded to only three orientations, which we refer to as mode 1, mode 2 and mode 3. The first starting orientation gave a single final orientation—“mode 1”. Starting orientations 2, 4, 5 and 6, being different in the beginning, gave the same final orientation with common amino acid patterns forming contact with lipids—“mode 2”. The starting orientation 3 gave a single final orientation—“mode 3”. These three modes were further investigated with full atom molecular dynamics.

After 200 nm full atom (FA) molecular dynamics simulation, all binding modes were stable. Final frames with protein at membrane surface and contact pattern analysis are shown on Suppl. Appendix A.

To further characterize the resulting binding modes, we estimated the protein altitude above the bilayer. The membrane interface (altitude 0) was calculated as an average position of phosphorus atoms. Such an estimate provides a direct connection with the AFM data. Analysis of protein altitude above the membrane surface by FA MD for three different modes gave altitude distributions centered at 1.74 nm (mode 1), 2.37 nm (mode 2), and 2.90 nm (mode 3) (Figure 4 and Appendix A). These were in good agreement with AFM data (Figure 2C), which also revealed three modes of bvPLA2 binding. Amino acid residues that form stable contacts with the membrane are listed in Appendix A for each binding mode. Mode 1 appeared to be the orientation with the highest contact intensity (highest number of amino acid residues in contact with the lipid molecules and highest number of contacts per frame).

Mode 1 had the biggest enzyme–bilayer contact spot and highest number of amino acid residues involved in the interaction with bilayer compared to mode 2 and mode 3 (Figure 3, bottom). The area of the enzyme–bilayer contact spot notably decreased from mode 1 to mode 3. We deduced that the energy of interaction decreased correspondingly from mode 1 to mode 3.

Enzyme surface regions involved in bvPLA2 interaction with bilayer partially overlapped or were in the close neighborhood. This phenomenon is highlighted in Figure 3, bottom. It is important that neighboring/overlapping regions belong either to mode 1/mode 2 or mode 2/mode 3 orientations, but not to the mode 1/mode 3 orientations. The last pair have no overlapping or neighboring regions.

According to altitude data, energy diagrams and sizes of the enzyme–bilayer contact spots, we assigned mode 1 found in MD to population 1 found by AFM, MD mode 2 to AFM population 2, and MD mode 3 to AFM population 3.

To find out if there was a difference in bvPLA2 structure in modes 1, 2 and 3, we performed cluster analysis of the protein. Cluster analysis revealed mobility (RMSD) of amino acid residues during simulation (Appendix A). In all modes, the highest mobility was observed in the 105–115 amino acids region—this is a part of the β-loop; while the mobility of this region decreased from mode 1 to mode 2, in mode 3 the β-loop appeared to be almost immobile. The reason for such fixation could be the interaction between β-loop amino acids and the lipid bilayer, especially between Arg108 and its neighbors (Appendix A).

## 3. Discussion

Membrane interaction is a necessary step for sPLA2 hydrolysis. Hydrogen bonds and polar and hydrophobic interactions are responsible for the formation of contact with the membrane. For each individual representative of sPLA2, the contribution of these forces is different. For bvPLA2, the role of charged amino acids is less important, and the contribution of aromatic and hydrophobic residues is stronger [27,28]. The surface of the enzyme that contacts the membrane is referred to as its interfacial binding surface (IBS). The interaction of the sPLA2 IBS with the membrane surface controls access of the lipid substrate to the active site and, therefore, the overall enzymatic activity. The orientation of the bvPLA2 on the bilayer surface was investigated by spin-labeling [29]. Incorporation of 13 spin labels into bvPLA2 showed preferable enzyme orientation and that bvPLA2 sits on the membrane surface rather than digging into the membrane.

Multimode binding. It is known that different sPLA2s are able to interact with structurally diverse molecules [4], suggesting that the (patho)physiological functions of sPLA2 are not limited to hydrolytic activity, and corresponding binding sites should exist. Heterogeneity of the sPLA2 behavior with the membrane was described by Gudmand et al. [30]. They succeeded in visualizing individual porcine pancreas sPLA2 molecules on a monolayer by fluorescence microscopy. The authors analyzed the trajectory of enzyme movement along the monolayer but not the enzyme globule. Two protein populations with different diffusion coefficients were visible—one moving fast and the other one moving slow. The majority of trajectories showed a combination of slow and fast diffusion steps [30]. Such behavior may indicate the existence of different binding modes for sPLA2. This aligns with our findings. Both our AFM and MD results revealed that bvPLA2 adapts three different orientations on the bilayer surface, which we defined as modes of binding. Below, we describe the observed binding modes and evidence for their existence according to data in the literature.

Mode 1. The mode 1 found in the present study matches with the previously described IBS of the bvPLA2. The amino acids of the IBS were estimated by X-ray structure [31], electron paramagnetic resonance study [29], and site-specific mutagenesis [32,33]. The molecular dynamic simulations of bvPLA2 resulted in the same list of amino acids in the IBS [15]. Amino acid residues that formed constant close contact with the membrane included the residues that form hydrogen bonds (Lys14, Agr23, Ser55, and Lys85) and salt bridges (Asp92 and Glu110) with lipid molecules, as well as aromatic (Tyr3, Phe24, Tyr81, Phe82, and Tyr134) and hydrophobic (Ile1, Ile2, Pro4, Ile78, Met86, and Ile91) residues providing protein accommodation on the membrane. bvPLA2 in mode 1 was the one most represented (up to 43% according to AFM data) and the one having highest contact intensity with the membrane (according to MD results).

Mode 2. The mode 2 is the second most frequent mode of binding with tryptophan residue (Trp128), which makes a notable contribution to the formation of the contact. In general, tryptophan plays an essential role in the interfacial binding and activity of different sPLA2s [27,28]. Typically, sPLA2s that contain tryptophan in IBS display the highest activity toward neutral phospholipid substrates [34,35], and the addition of tryptophan to IBS significantly enhanced the overall enzymatic activity [36]. Protein binding to the bilayer surface could be traced by intrinsic protein fluorescence if the protein has tryptophan residues involved in the binding. The method was used several times to trace sPLA2 binding to the bilayer surface [33,37]. In our previous work, we traced changes in bvPLA2 intrinsic fluorescence upon POPC addition [15,38]. It is important that there is no tryptophan residue in the IBS of bvPLA2 (Figure 3, Appendix A) [15]. On the other hand, Trp128 residue is among the amino acids that form enzyme–bilayer contact in mode 2. We assume that known changes of tryptophan fluorescence upon bvPLA2 binding to bilayer surface may be due to the mode 2 contribution. In recent work, Nasri et al. investigated the role of Trp128 in bvPLA2 activity [39]. The authors converted Trp128 to N’-formylkynurenine product by the action of singlet oxygen (remaining protein staying intact). The modification reduced lipid hydrolysis by 80%, that is, this residue is important for the functioning of bvPLA2. Since Trp128 is outside both catalytic and IBS sites, the results by Nasri et al. could be explained by the existence of mode 2. We can speculate that formylkynurenine, being a reactive species, can react with an amino group of DOPE lipid (involved in the experiments by Nasri et al.), thus stabilizing mode 2 and depopulating mode 1, preventing hydrolysis.

Mode 3. In mode 3, enzyme contacts the bilayer through the β-loop. The role of the β-loop of bvPLA2 in supporting interfacial catalysis was examined with the D99-118 deletion mutant by Gromashchi et al. [32]. This mutant refolded with about half the yield obtained for WT and displayed kinetic and vesicle binding properties virtually identical to those of WT. All secreted PLA2s have this β-loop, but its function has not been identified [32]. In the case of bvPLA2, the β-loop does not play a significant role in interfacial binding and catalysis on anionic interfaces, but we assume that it may play a role in product release to the milieu (see below). Mode 3 has the smallest contact area with the membrane (and corresponding energy of interaction), and the desorption of the enzyme from the membrane is most probable. Thus, switching to mode 3 may be important for activating hopping of the enzyme from bilayer to bilayer.

Switching modes. According to energy differences between modes (Figure 2C), the enzyme could relatively easily switch between mode 1 and mode 2. The mode 2—mode 3 transition costs two-fold more energy than mode 1—mode 2, thus it is less probable. We suggest that mode 1—mode 3 transition should come through an intermediate step at mode 2, as direct transition is energetically unfavorable. The same result was deduced from MD data. The enzyme–bilayer contact spots depicted in Figure 3 assume that transition from mode 1 to mode 3 does include an intermediate step at mode 2.

The existence of multiple modes of binding has biological significance. These binding modes can be interpreted as intermediate steps of sPLA2 movement along the bilayer surface with binding lipids and releasing products (Figure 5). Switching between modes 1 and 2 changes the contact spot; while lipids that were in contact disperse, neighboring lipids take their place, and enzyme moves from one lipid to another. After the round of catalysis, products may be released into solution or be retained in the membrane. Individual components of hydrolysis products—fatty acids and lyso-phospholipids—are poorly soluble in water; when mixed together at 1:1 mol. ratio they form a stable bilayer [40]; when added to lipids in different proportions, they also form a bilayer [21]. We propose a hypothesis about the release of hydrolysis products through a change in the mode of protein binding to the membrane. If the enzyme is in mode 1 or mode 2 after the hydrolysis cycle, the products are released into the bilayer. Alternatively, switching to mode 3 facilitates product release into the milieu. Release of free fatty acid, especially arachidonic acid, underlies the regulation of inflammatory processes with the participation of sPLA2. In general, the kinetic model of sPLA2 hydrolysis implies both the release of the fatty acid into the medium and its incorporation into the original lipid aggregate [1,3]. As was shown in vitro, sPLA2s can release arachidonic acid by two mechanisms: an external plasma membrane pathway and a heparan sulfate proteoglycan-shuttling pathway [3,41,42]. The switching to mode 3 may explain how the products of the reaction could be released into a milieu or presented to a carrier. The ratio of protein populations in different binding modes may vary depending on the lipid composition of the membrane. Despite high homology of different representatives of sPLA2, the behavior shown on the membrane should be specified for each individual protein.

## 4. Conclusions

Here, we reported that the lipid membrane is involved in the catalytic cycle of a peripheral protein, phospholipase A2, by accommodating three different orientations of the enzyme. The protein has three different membrane binding modes. This finding was obtained independently in “wet” experiments using atomic force microscopy and in in silico experiments using molecular dynamics simulations. Our main observation was that sPLA2 could bind the bilayer in different modes, but not just in a single one as assumed previously. The existence of multiple modes of binding may have biological significance. We suggest that switching between modes 1 and 2 is responsible for the enzyme moving along the membrane; mode 3 is important for both product release in the biological milieu and enzyme desorption from the bilayer surface.

## 5. Materials and Methods

Palmitoyloleoylphosphatidylcholine (POPC) was purchased from Avanti Polar Lipids (USA); Phospholipase A2 from bee venom was obtained from Sigma-Aldrich. Enzyme stock solution (1.0 mM) was prepared in 0.1 M Tris-HCl buffer containing 0.1 M NaCl, pH 8.5. Enzyme concentration was controlled by the absorption at 280 nm (Nano Drop OneC, Thermo Fisher). 2-Amino-2-(hydroxymethyl)propane-1,3-diol (Tris base, Fluka), HCl, NaOH, ethylenediaminetetraacetic acid (EDTA, Panreac), and CaCl_2_ were of reagent grade.

### 5.1. Liposome Preparation

Liposomes were prepared by lipid film hydration and sonication. Briefly, an aliquot of POPC was evaporated from chloroform in a round-bottomed flask under N_2_ stream. The lipid film was hydrated for 1 h at room temperature in 100 mM NaCl. The suspension was then sonicated for 20 min in ultrasound bath. Usually, liposomes were prepared at a concentration of 2 mg/mL.

### 5.2. Atomic Force Microscopy (AFM) Study

#### 5.2.1. Bilayer Formation on the Mica Surface

Freshly prepared liposomes were applied to the surface of freshly cleaved mica (100 μL 0.25 mg/mL), incubated for 2 to 5 min (depending on the experiment, to obtain mica partially or fully covered with bilayer). Excess liposomes were removed by washing, and supported bilayer was rinsed at least 5 times with 10 mM Tris-HCl buffer containing 100 mM NaCl, pH 8.5, with or without 1–1.5 mM EDTA (or 5 mM CaCl_2_).

#### 5.2.2. bvPLA2 Absorption

After bilayer formation on the mica surface, 100 μL of bvPLA2 solution (1–0.5 μM for inhibited enzyme, 0.025 μM for active enzyme) was added for 10 min, then rinsed off with the same buffer to remove unbound enzyme. Samples were imaged in tapping mode using a Multimode Nanoscope V AFM (Bruker) equipped with the J type scanner and an electrochemical fluid cell. To obtain high resolution, ultrasharp SNL-10 cantilevers with a nominal spring constant of 0.06 N/m and a tip radius of approximately 2 nm (Bruker) were used. Scan rate was typically 2 Hz. Image processing was performed using the FemtoScan Online software (Advanced Technologies Center).

#### 5.2.3. Data Analysis

To characterize bvPLA2 globule on POPC supported bilayer, we analyzed the mean altitude above bilayer for collected images. The distribution of altitude Z along the sample was obtained through Bruker software associated with the instrument. Individual protein globule on bilayer surface was analyzed, and data were collected from at least three independent experiments, each including 2–10 fields of view with up to 100 protein globules. The data were than fitted using a sum of Gaussian functions. The math workup was performed using Python (NumPy) built-in routines.

### 5.3. Molecular Dynamics

The MD simulations were performed according to our previously published methods [15], with minor modifications. The protein structure was obtained from PDB (PDB ID: 1POC [31]); transition-state analogue was removed.

In the first step, we modeled PLA2 landing on the bilayer surface. The modeling was performed using the CG approach. The protein was introduced ~1 nm above the pre-balanced membrane (15 × 15 lipids), then water grains (W), antifreeze (WF), and 2 Cl− grains were added to neutralize the protein charge. The protein was oriented in one of the six positions corresponding to six sides of the cube, where we virtually placed the protein. The size of the starting system was 12 × 12 × 15 nm. The resulting system was balanced by energy minimization and two consecutive MD calculations with a duration of 1 ns and an integration step of 2 and 5 fs. Then, a 5 μs CG MD simulation with an integration step of 20 fs was performed. The calculations were performed in the Gromacs program using the Martini 2.2 force field for protein and standard lipid topologies [43]. To maintain protein structure when creating its topology, the rubber bond scheme [44] was used. The simulation was carried out at 295 K. In the calculations, the parameters recommended for calculating protein–membrane systems with Martini 2.2 force field (‘new’ version) [45] were used, and the integration step was reduced to 20 fs to retrieve correct dynamics of the aromatic groups of the protein. To analyze protein–membrane interactions, we developed a utility that searches for pairwise contacts (nuclei with a distance of less than 0.6 nm) between the grains of protein and lipids along the entire trajectory (Appendix A).

In the second step, orientations of bvPLA2 that resulted in the different contact patterns with lipid molecules in CG dynamics were refined in full-atom MD (Appendix A). The conversion from coarse-grained to full-atom representation was carried out using an in-house developed utility. The position of the protein in the full-atom system was obtained by fitting C α atoms on BB grains. An initial three-dimensional structure of bvPLA2 was obtained from PDB, as during the first step of modeling, the protein structure changed insignificantly (RMSD < 3 Å). To maintain the protein structure in the active conformation, a Ca^2+^ ion was placed in the active center of the enzyme and its position was maintained by distance limitations with the polar groups that coordinated the ion. Then, water molecules and ions were added to the system. The resulting system was balanced by energy minimization and 1 ns MD simulation with restrictions on the position of heavy atoms of the protein and integration step of 1 fs. Next, a 200 ns simulation was performed. Gromacs molecular dynamics package version [46] with Amber-14SB force field parameters for proteins [47], Slipids parametrization of lipid molecules [48], and TIP3P water model [49] was used. In all MD simulations, protein, lipids, and solvent (water and ions) were coupled by Nose–Hoover thermostat separately, and temperature was maintained at 295 K. A constant pressure of 1 atm was controlled by the Berendsen barostat in semi-isotropic mode. Long-range interactions were calculated using the cutoff algorithm with the limit of 1.4 nm for van der Waals interactions and using the PME scheme with Fourier spacing of 0.12 nm for electrostatic interactions. For the obtained trajectories, we analyzed protein RMSD, the number of protein contacts with the membrane (contacts between heavy atoms), and protein altitude above membrane surface (interface was estimated as average position of the phosphorus atoms of the POPC molecule during simulation).

## Figures and Tables

**Figure 1 toxins-14-00669-f001:**
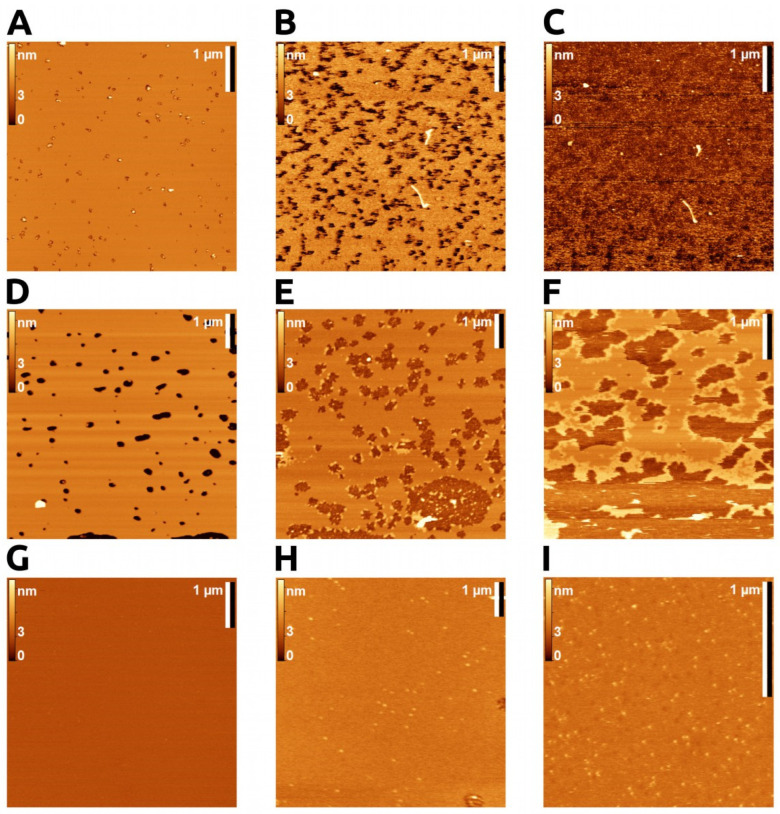
AFM imaging. POPC bilayer topography upon bvPLA2 addition in hydrolytic and non-hydrolytic conditions. (**A**–**C**) Active enzyme on POPC bilayer; (**A**) smooth membrane, almost full coverage, before enzyme addition. (**B**) The same area 2 min after addition of bvPLA2 (0.01 μM with 5 mM CaCl_2_; rapid membrane degradation started from the bilayer defects. (**C**) The same area 20 min after enzyme action, with complete bilayer degradation. (**D**–**G**). Inhibited enzyme (with 25 mM EDTA). (**D**) POPC bilayer with small areas of bare mica (dark spots). (**E**) Addition of bvPLA2 (1 μM with 25 mM EDTA; enzyme tended to adsorb on the membrane boundaries and on bare mica areas. (**F**) Clusters of bvPLA2 at the edge of membrane, with no enzyme on the membrane surface (high EDTA concentration hampered enzyme adsorption). (**G**–**I**) Inhibited enzyme visualized on membrane surface. (**G**) Total coverage of mica by POPC membrane. (**H**,**I**) bvPLA2 visualized on POPC membrane (0.5 μM bvPLA2 with 5 mM EDTA and 1 μM bvPLA2 with 1 mM EDTA), correspondingly.

**Figure 2 toxins-14-00669-f002:**
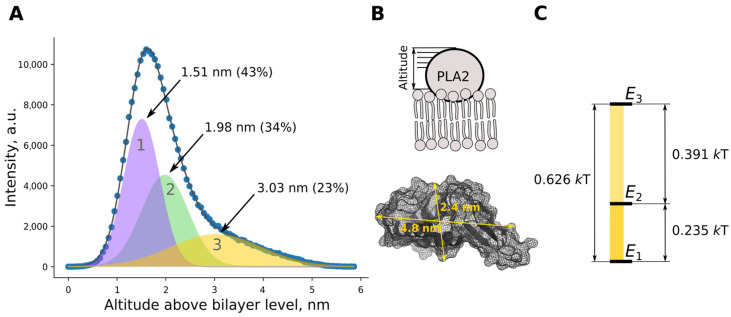
(**A**) Combined size distribution of bvPLA2 molecules on the surface of POPC bilayer obtained by AFM image analysis (circles). Fit of experimental data with a sum of Gaussian functions (gray solid line). Gaussian functions obtained through fitting are represented as filled curves. In total, three different modes of enzyme altitude above the bilayer were identified upon fitting experimental data. Digits represent median and fraction values of each Gaussian function. (**B**) Schematically, altitude is the distance between membrane surface and top surface of the enzyme. Distances Asp69—His11 (vertical rule) and Glu20—Thr103 (horizontal rule) give an idea of bvPLA2 dimensions. (**C**) Energy difference between binding modes calculated according to Boltzmann equation.

**Figure 3 toxins-14-00669-f003:**
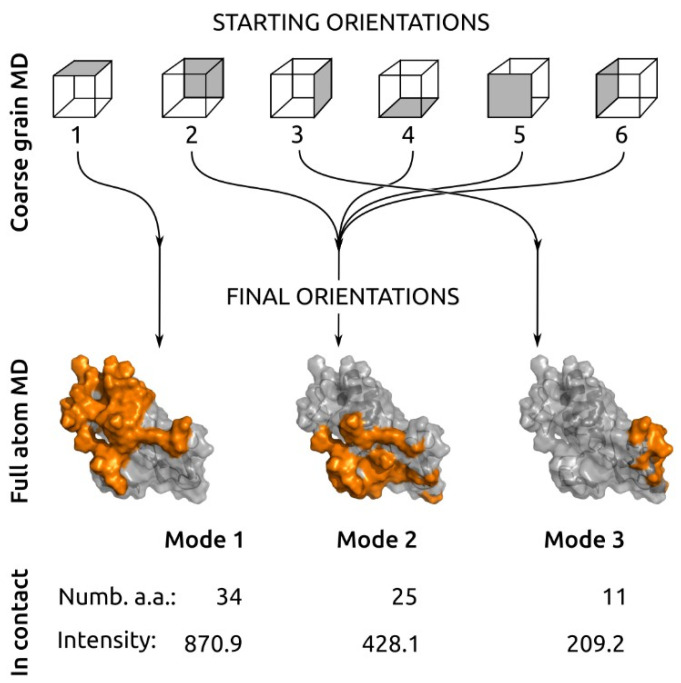
**Top**: Schematic representation of six starting orientations of bvPLA2 with respect to bilayer surface in coarse grain MD. At the end of simulation, these degraded to only three different modes. **Bottom**: Full atom MD for three binding modes of bvPLA2 on POPC bilayer. View from the membrane surface in the “mode 1” orientation. Amino acid residues that form stable contacts with lipids are highlighted in orange with their total number and contact intensity. For more details, see Appendix A.

**Figure 4 toxins-14-00669-f004:**
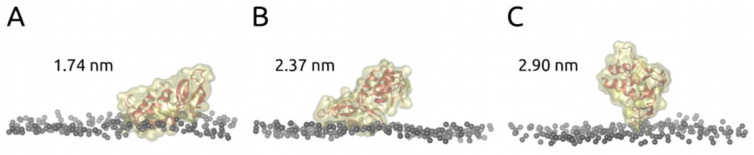
Full atom MD final frames for 200 ns simulation of bvPLA2 on POPC bilayer. The average height of the protein above the bilayer surface (relative to the average position of the phosphorus atoms of the POPC) is indicated for each of the found stable binding modes (mode 1 (**A**), 2 (**B**) and 3 (**C**)). For more details, see Appendix A.

**Figure 5 toxins-14-00669-f005:**
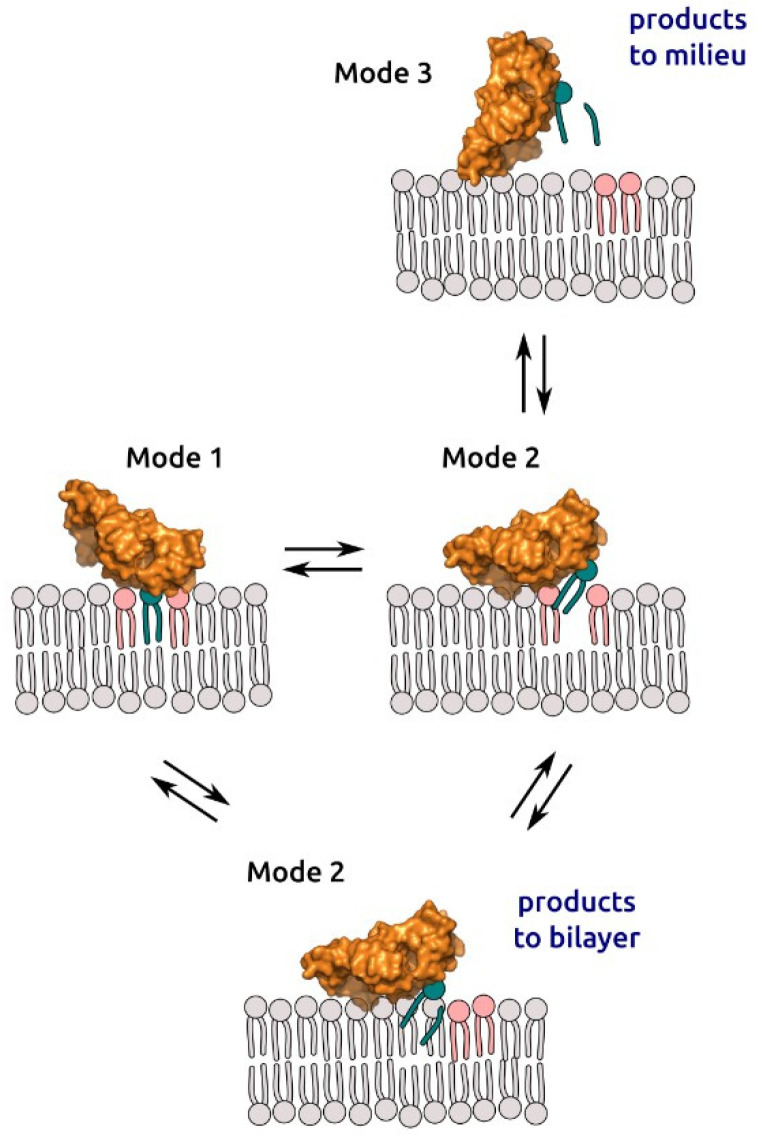
Proposed model of sPLA2 switching binding modes during the hydrolysis process. Lipids colored in pink represent lipids at the initial contact spot. Constant switching between mode 1 and mode 2 resembles see-sawing, but the motion is more complex since the enzyme moves along the bilayer surface. Enzyme, while see-sawing, pulls a lipid from the bilayer, hydrolyzes it and releases products back to the bilayer or to the milieu.

## Data Availability

Not applicable.

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
