# Peer review of "sPLA2 Wobbles on the Lipid Bilayer between Three Positions, Each Involved in the Hydrolysis Process"

_toxins, 2022, doi:10.3390/toxins14100669_

Round 1
Reviewer 1 Report
Using atomic force microscopy, the authors demonstrated that lipid membranes are involved in the catalytic cycle of peripheral PLA2 by responding to three different orientations of the enzyme for its interaction with the cell membrane using bee PLA2. Proven in 'wet' experiments and in silico experiments using molecular dynamics.
As a reviewer, we would like to request that the limitations of this study be described in the discussion. Also, as a correction item in the text, D in Fig. 2 is not written, so we would like to add it.
Author Response
Point 1: As a reviewer, we would like to request that the limitations of this study be described in the discussion. Also, as a correction item in the text, D in Fig. 2 is not written, so we would like to add it.
Response 1: Limitations of the provided hypotheses and discussion on its applicability was added to the manuscript (see lines 306-309).
Point 2: Also, as a correction item in the text, D in Fig. 2 is not written, so we would like to add it.
Response 2: Thank you for indicating to this mismatch. Indeed, reference to Figure 2, D was incorrect, now we changed it to correct indication - Figure 2, C (see line 140).
Reviewer 2 Report
Article "sPLA2 wobbles on lipid bilayer between three positions, each involved in hydrolysis process" is an excellent work on the dynamics of the action of bee venom PLA2 on membranes. The paper is well elaborated, the illustrations adequate and the message well explained in the discussion and conclusions.
Phrases and their initials should be revised. Name them the first time and then continue with their initials. Review for example DPPC, POPC, AFM, bvPLA2.
Author Response
Point 1: Phrases and their initials should be revised. Name them the first time and then continue with their initials. Review for example DPPC, POPC, AFM, bvPLA2.
Response 1: We thank reviewer for close reading of our manuscript and high appreciation of our work. All abbreviations was defined the first time they appeared in the abstract and the main text, unnecessary abbreviations was removed. Namely: Lp-PLA2 was changed to lipoproteine-associated PLA2 (line 42); DPPC was replaced by dipalmitoilphosphatidylcholine (lines 45, 46, 116); abbreviations for EDTA (line 82), SLB (line 80), bvPLA2 (line 69), POPC (lines 70, 331) were defined.